# Design and Synthesis of Pyrrolidinyl Ferrocene-Containing Ligands and Their Application in Highly Enantioselective Rhodium-Catalyzed Olefin Hydrogenation

**DOI:** 10.3390/molecules27186078

**Published:** 2022-09-17

**Authors:** Xin Li, Therese B. Brennan, Cian Kingston, Yannick Ortin, Patrick J. Guiry

**Affiliations:** 1Centre for Synthesis and Chemical Biology, School of Chemistry, University College Dublin, Belfield, D04 N2E5 Dublin, Ireland; 2MSD Ballydine, Kilsheelan, Co. Tipperary, E91 V091 Clonmel, Ireland; 3Synthesis and Solid State Pharmaceutical Centre, School of Chemistry, University College Dublin, Belfield, D04 N2E5 Dublin, Ireland

**Keywords:** chiral ligand, ferrocene, asymmetric catalysis, rhodium-catalyzed olefin reduction

## Abstract

Herein, we report the design and synthesis of a series of chiral pyrrolidine-substituted ferrocene-derived ligands. The proficiency of this novel structural motif was demonstrated in the Rh-catalyzed asymmetric hydrogenation of dehydroamino acid esters and α-aryl enamides. The products were obtained with full conversions and excellent levels of enantioselectivities of up to >99.9% ee and 97.7% ee, respectively, using a BINOL-substituted phosphine-phosphoaramidite ligand which possesses planar, central, and axial chirality elements.

## 1. Introduction

Phosphorous-based ligands have found extensive use in homogeneous transition metal catalysis [1,2,3]. Bidentate P,P based structures, exemplified by BINAP, DuPhos and Josiphos, represent a privileged ligand scaffold in transition-metal catalysis (Figure 1) [4]. Ferrocenyl-based Josiphos derivatives have proven to be efficient ligands across a range of enantioselective processes, particularly in the field of olefin hydrogenations [5,6]. Their value has been demonstrated in the large-scale Ir-catalyzed hydrogenation of an N-aryl imine in the synthesis of the herbicide (*S*)-metolachlor [7]. Thus, the increasing demand for efficient enantioselective technologies has led to significant investigation of ligand derivatives, as evidenced by the range of commercially available variants and sub-families such as Knochel’s Ferriphos ligands [8].

Focusing on planar chiral ferrocene compounds and their ease of preparation through the use of diastereoselective *ortho*-metalating groups, including amines, sulfoxides, acetals, oxazolines, azepines, sulfoximines, and hydrazones, has contributed to the wide range of planar chiral ferrocene ligands reported to date [9]. We reported the preparation of ferrocenylphosphinamine ligands of type **1**, Figure 2, possessing both planar and central chirality obtained through diastereoselective metalation of *trans*-(2*R*,5*R*)-2,5-dialkyl-1-(ferrocenylmethyl)pyrrolidines and their application in Pd-catalysed allylic alkylation [10].

Thus, interested by the successful application of ferrocene-containing ligands, bearing an α-chiral center, and extending our work on pyrrolidine-containing P,N ligands [11,12,13], we developed novel ferrocene ligands in which the α-chiral center is incorporated into a pyrrolidine unit. We had previously reported the enantioselective preparation of ferrocenepyrrolidine (*R*)-**2** and applied it in the diastereoselective formation of a series of N,O ligands of type **3** for the diethylzinc-mediated addition to aldehydes, affording enantioselectivities of up to 95% ee [14]. In addition, we reported the synthesis of novel ferrocene-phosphinamine ligands of type **4**, again obtained through diastereoselective *ortho*-metalation of ferrocenepyrrolidine (*R*)-**2**, Figure 2, and their application in the Pd-catalyzed allylic alkylation of 1,3-diphenylprop-2-enyl acetate with dimethyl malonate gave enantioselectivities of up to 77% ee [15].

In 2002, Boaz reported the synthesis of air-stable ‘BoPhoz’ ferrocenylphosphine-aminophosphine ligands (**5**, Figure 3), which were highly effective in the asymmetric hydrogenation of dehydro—α-amino acids, itaconic acids, and α-ketoesters [16,17]. Two different phosphorus donor atoms generate electronic asymmetry at the metal centre which then provides unique modulation of the catalyst activity. Derivatization of the ligands led to an expansion of the range of suitable substrates for hydrogenation [18,19,20] and identified the proficiency of the ligands in the catalytic asymmetric synthesis of cyclohexenone-based atropisomers [21]. Chang and Zheng introduced a phosphine-phosphoramidite scaffold (**6**, Figure 3) which proved effective in the asymmetric hydrogenation of a broad range of substrates including both (*Z*)- and (*E*)-aryl and β-alkyl-β-(acylamino)acrylates [22,23,24]. With the success of the reported ligand derivatizations in mind, it occurred to us that the flexible amino sidechain in **5** and **6** could be modified to introduce the more rigid pyrrolidinyl motif found in **3** and **4**. Herein, we present the preparation of a variety of ferrocenylphosphine-aminophosphine and ferrocenylphosphine-phosphoramidite ligands of type **7** and **8**. The relatively facile introduction of the amino-phosphine moiety introduces a modular element, which enabled the preparation of a diverse range of novel ligands for investigation.

## 2. Results and Discussion

### 2.1. Synthesis and Characterization of N-Phosphorus-Substituted Pyrrolidine Based Ligands

The strategy for the preparation of ferrocenyl N-phosphinepyrrolidinyl ligands was adapted from our previous work on the ferrocene-phosphinamine ligands of type **4** [15]. The synthesis of pyrrolidine (*R*)-**13** from ferrocene (**9**), consisted of a Friedel-Crafts acylation to afford ketone (**10**), a Corey-Bakshi-Shibata oxazoborolidine-mediated enantioselective reduction to afford alcohol (*R*)-**11**, its acetylation to afford acetate (*R*)-**12** and finally treatment with allylamine to give (*R*)-**13** in a 92% yield over four steps (Figure 1).

The introduction of the ferrocenyl phosphine moiety was accomplished by non-selective *ortho*-lithiation of (*R*)-**13,** which was then quenched with chlorodiphenylphosphine (Figure 2). Diastereomers (*R*,*Sp*)-**14** and (*R*,*Rp*)-**14** were separated by silica gel column chromatography and isolated in yields of 24% and 33%, respectively. To facilitate the introduction of the N-pyrrolidinyl phosphine substituent, deallylation of the amine was accomplished using palladium(tetrakistriphenylphosphine) and *N*,*N*-dimethyl barbituric acid (NDMBA) [25]. Deprotected pyrrolidines (*R*,*Sp*)-**15** and (*R*,*Rp*)-**15** were isolated in yields of 98% and 86%, respectively. With the synthetic precursors to the desired ligands in hand, the final coupling could now be performed using triethylamine and the appropriate chlorodiarylphosphine. The initial series of ligands **L1**–**6** were synthesized using (*R*,*Sp*)-**15**. Ligands bearing neutral (**L1**), electron rich (**L2**), and electron poor (**L3**–**L4**) aryl groups were synthesized in moderate to good yields (40–78%). The proficiency of a BINOL unit in ferrocenyl bisphosphonate [26], phosphoramidite [27,28], and phosphine-phosphoramidite ligands [24] for enantioselective rhodium-catalyzed hydrogenation has been well-documented. Therefore, phosphoramidites **L5** and **L6** were synthesized using (*R*)- or (*S*)-1,1′-binaphthyl-2,2′-diyl phosphorochloridate in good yields of 83 and 70%, respectively. In order to test the effect of a switch in planar chirality, isomeric ligands **L7** and **L8** were synthesized from (*R*,*Rp*)-**15** in excellent yields of 90% and 98%, respectively (Figure 4).

Interestingly, ^5^*J* coupling between the phosphorus atoms was observed for ligands **L1**–**L6** with coupling constants ranging from 16.4 to 56.2 Hz. Cross peaks in the two-dimensional ^31^P{^1^H}–^31^P{^1^H} spectra (^31^P COSY) provided further evidence for this unexpected interaction (Figure 5). Although ‘long range couplings’ (across more than four bonds) between two phosphorus atoms are quite rare, the phenomenon has been observed with several ligands, such as Xantphos (^6^*J* coupling) and tetraphosphine ferrocenyl derivatives [29]. An excellent in-depth study of the ferrocenyl compounds attributed the ^31^P–^31^P nuclear spin-spin coupling to a through-space non-bonded interaction of the phosphorus lone pairs, as was previously observed in ^19^F–^19^F and ^15^N–^19^F couplings [30]. Due to the magnitude of the coupling constants observed for **L1**–**L6**, it is unlikely a through-bond interaction (*σ*- and *π*-transmitted components) is taking place. Through-space coupling results from overlap of the phosphorus lone-pair orbitals. Although the interaction provides an adequate pathway to transmit spin information between the nuclei, it does not lead to chemical bonding because both orbitals are occupied.

A comparison of the *J*_PP_ coupling constants in **L1**–**L8** revealed a considerable effect resulting from the choice of aryl phosphine-substitution (Table 1). The magnitude of the coupling constant depends on the extent of the lone-pair overlap, providing some information on the P-P orientation and proximity in solution. No coupling was observed for **L7**–**L8** indicating the planar and central chirality of the ligand prohibit favorable alignment of the lone pairs. These factors could provide useful information on the bite angle and play an important role in the future development and application of this class of ligand.

X-ray crystallographic analysis of **L5** provided confirmation of the assignment of the planar, central, and axial chirality in the molecule; 15% oxidation of the ferrocenyl phosphorous atom (P1) was observed (Figure 6). The distance between the phosphorus atoms was measured at 3.9 Å, which favorably compared to Meunier’s observation of coupling constants over 20 Hz with P–P distances of 4.0 Å or below [30]. However, the solid-state conformation of **L5** obtained from X-ray structural data should not reflect the conformation in solution due to effects of crystal-packing, meaning the true inter-phosphorus distance remains unknown. 

#### 2.1.1. Rhodium-Catalyzed Asymmetric Hydrogenation of Dehydroamino Acid Esters

Transition metal-catalyzed asymmetric hydrogenation of dehydroamino acid derivatives is a valuable method for the preparation of amino acid precursors [31]. As such, the field has undergone extensive investigation since the 1970s, from the seminal work of Knowles and Kagan [32,33,34], thereby providing a convenient test reaction to test the efficiency of the rhodium complexes of our novel ligands **L1**–**L8.**

##### Reaction Condition Optimization

(*Z*)-Methyl-2-acetamido-3-phenylacrylate **16a** was chosen as a suitable substrate to test the application of ligands **L1**–**L****8** (Table 2). Initial conditions consisted of Rh(COD)_2_OTf as the rhodium source, with ligand **L1** in THF at room temperature under an atmosphere of hydrogen for 12 h. Full conversion of the starting material was observed with an *ee* of 89.0 % *ee* for the product (Table 2, entry 2). When Rh(COD)_2_BF_4_ was employed a drop in the conversion of starting material was observed, although the product was formed in a similar *ee* (Table 2, entry 3). This effect was subsequently observed throughout the optimization process (Table 2, entries 11, 18). Increasing the hydrogen pressure was found to have a detrimental effect upon the *ee* of the product (Table 2, entries 4-5). Similarly, variation of the solvent had a deleterious effect upon the *ee* and in some cases, the conversion (Table 2, entries 6–10). Poor asymmetric induction and conversion were observed upon switching to ligand **L2,** which bears an *o*-tolyl substituted phosphine (Table 2, entries 12–13). The comparatively electron-poor *p*-fluoro substituted **L3** gave a similar result to that obtained previously with ligand **L1** (Table 2, entry 14). Switching to 3,5-di(trifluoromethyl)-substituted ligand **L4**, the product was formed in 92.2% *ee* with full conversion after 2 h, and the *ee* was further increased to 95.5% upon cooling the reaction mixture to 0 °C for 3 h (Table 2, entries 15–16). An (*R*)-BINOL-based system was next evaluated in the application of ligand **L5**. Gratifyingly, full conversion of **16a** was found in 12 h, with the product formed in an excellent enantiomeric excess of 97.9% (Table 2, entry 18). In this case, upon repetition with Rh(COD)_2_BF_4_, no decrease in the conversion of the starting material was observed (Table 2, entry 19). By increasing the hydrogen pressure to 10 bar, full conversion of the starting material was observed within 2 h, but the *ee* of the product dropped slightly (Table 2, entry 20). Variation of the solvent did not have a beneficial effect upon the enantioselectivity of the reaction (Table 2, entries 21–22). Interestingly, changing the axial chirality using (*S*)-BINOL in ligand **L6** resulted in a significantly lower *ee* for the product (Table 2, entry 23). The best result was obtained by switching the planar chirality, as illustrated in ligand **L7,** where full conversion of the starting material was observed in one hour and an *ee* of 99.5% was obtained (Table 2, entry 24). By increasing the hydrogen pressure to 20 bar and the reaction time to 4 h, the catalyst loading could be significantly lowered to only 0.02 mol %, with minimal effect on the enantioselectivity of product formation (Table 2, entry 25). Interestingly, in contrast to the planar isomers (ligand **L5** and **L6**), when the opposite hand of BINOL was used with ligand **L8,** the opposite enantiomer of the product was formed, albeit with a much lower *ee* of 46.7% (Table 2, entry 26). For comparison purposes, the results obtained in the literature using rhodium complexes of Josiphos, DuPhos, and BoPhoz ligands are included (Table 1, entries 27–29). Our optimal ligand (99.5% ee) compares favorably with DuPhos (85% ee) and Josiphos (96% ee) and is identical to the level of enantioselectivity exhibited by the BoPhoz ligand (99.5% ee). 

##### Substrate Scope

With optimum conditions in hand, the activity of a selection of ligands was investigated with a variety of amino acid precursors (Figure 3). As the best ligand from the optimization process, **L7** consistently gave excellent results of full conversion and over 99% *ee* regardless of any electron-rich (**16b**, **16c**, **16j**) or electron-poor substituents (**16d**–**16i**) on the β-aryl moiety or the particular substitution pattern of the substrate. The excellent performance also extended to a dehydroamino acid ester without aryl substitution (**16k**). Phenyl-substituted ligand **L1** was also tested across a selection of the substrates and gave consistent levels of asymmetric induction (88.5–92.0% *ee*) but variable conversions of the starting material from (60–>99%). The rhodium complex of ligand **L4** bearing 3,5-di(trifluoromethyl)phenyl substituents also led to product formation, with full conversions and remarkably consistent, although lower, enantioselectivities regardless of the starting material (92.5–96.3% *ee*).

#### 2.1.2. Rhodium-Catalyzed Asymmetric Hydrogenation of α-Aryl Enamides

With the success of the rhodium complexes in the highly enantioselective reduction of dehydroamino acids, the efficiency of the ligands was next tested in the rhodium-catalyzed hydrogenation of a selection of α-aryl enamides. This is a valuable process for the construction of a variety of amines. 

##### Reaction Condition Optimization

*N*-(1-phenylvinyl)acetamide **18a** was chosen as the model substrate for optimization studies. Ligand **L1** was tested using Rh(COD)_2_OTf in THF at room temperature with 40 bar hydrogen pressure for 2 h. While full conversion of the starting material was observed, a disappointing *ee* for the product of 33.6% (*S*) was observed (Table 3, entry 2). Decreasing the hydrogen pressure did not have a significant effect on the level of asymmetric induction (Table 3, entry 3). Shortening the reaction time to 1 h and switching to MeOH as solvent provided a slight increase to 46.0% *ee* (Table 3, entry 5). While ligands **L2** and **L3** were similarly ineffective, a dramatic improvement was observed upon using the 3,5-di(trifluoromethyl)-substituted ligand **L4** with an *ee* of 91.3% (Table 3, entries 6–8).

Switching to ligand **L5** in THF at 10 bar hydrogen provided the product with similar levels of asymmetric induction (Table 3, entry 9). Decreasing the pressure further to 1 bar resulted in decreased conversion of the starting material (Table 3, entry 10). As in our previous optimization, the best result was observed with ligand **L7,** with an *ee* of 96.4% observed and full conversion after 2.5 h (Table 3, entry 11). A similar *ee* was observed with 10 bar hydrogen pressure after 1 h while decreasing the catalyst loading to 0.2 mol % provided the product in 91.0 % *ee* (Table 3, entries 12-13). Once again, switching the axial chirality of the BINOL moiety provided the opposite enantiomeric product in a much-lowered *ee* of 31.7 % (Table 3, entry 14). For comparison purposes, the results obtained in the literature using rhodium complexes of Me-BPE (related to DuPhos) and BoPhoz ligands are included (Table 2, entries 15–16). Our optimal ligand (96.4% *ee*) compares closely with Me-BPE (95.2% *ee*) but outperforms the level of enantioselectivity exhibited by the BoPhoz ligand (61.8% *ee*).

##### Substrate Scope

The optimized conditions with ligand **L7** were tested across a range of α-aryl enamides (**18a**–**f**) and, in contrast to the investigation of dehydroamino acids, the results were substrate-dependent (Figure 4). Naphthyl (**18b**) and 4-chloro (**18c**) substitution of the aryl ring resulted in full conversions and excellent enantioselectivities (97.4–97.7% *ee*), but a lower *ee* of 93.7% was observed with a 4-methoxy substituent (**18d**). Ligand **L4** was also tested in the synthesis of **19b** and **19c** and the products were formed in slightly lower enantioselectivities (92.0–92.4% *ee*), while a significant drop in the level of asymmetric induction was seen for the formation of **19d** (77.6% *ee*). Poor conversion of the starting material and low level of asymmetric induction were observed in the hydrogenation of bicyclic *N*-(3,4-dihydro-1-naphthyl)acetamide **18e** with ligand **L7** (23% conversion, 48.0% *ee*) and ligand **L4** (12% conversion, 27.0% *ee*). A similarly poor result was observed in the hydrogenation of β-phenyl- β- (acylamino)acrylate **18f** with ligand **L7** (18% conversion, 15.3% *ee*). However, a drastic improvement to full conversion to the product with an *ee* of 61.3% was observed with ligand **L1**. The result demonstrates the need for substrate dependent optimization with a new class of substrate to maximize the potential of the ligand series developed.

## 3. Materials and Methods—Chemistry

Unless otherwise noted, reactions were performed with rigorous exclusion of air and moisture under an inert atmosphere of nitrogen in flame-dried glassware with magnetic stirring using anhydrous solvents. N_2_-flushed stainless steel cannulas or plastic syringes were used to transfer air and moisture-sensitive reagents. All reagents were obtained from commercial sources and used without further purification unless otherwise stated. All anhydrous solvents were obtained from commercial sources (Sigma Aldrich, Glasgow, United Kingdom) and used as received with the following exceptions: diethyl ether (Et_2_O), dichloromethane (CH_2_Cl_2_), and toluene (PhCH_3_) were dried by passing through activated alumina columns. Powdered activated 4 Å molecular sieves were purchased from Sigma Aldrich (Glasgow, United Kingdom)and were stored in an oven at 120 °C. In vacuo refers to the evaporation of solvent under reduced pressure on a rotary evaporator. Thin-layer chromatography (TLC) was performed on aluminium plates pre-coated with silica gel F254 (Merck, Darmstadt, Germany). They were visualised with UV-light (254 nm) fluorescence quenching, or by charring with Hanessian’s staining solution (cerium molybdate, H_2_SO_4_ in water), basic potassium permanganate staining solution (potassium permanganate, K_2_CO_3_ and NaOH in water), or an acidic vanillin staining solution (vanillin, H_2_SO_4_ in ethanol). Flash column chromatography was carried out using 40–63 μm, 230–400 mesh silica gel.

^1^H NMR spectra were recorded on a 300-, 400-, or 500-MHz spectrometer. ^13^C NMR spectra were recorded on a 400- or 500-MHz spectrometer (Agilent, Birmingham, United Kingdom) at 101 or 126 MHz. ^19^F NMR spectra were recorded on a 400-MHz spectrometer at 470 MHz. Chemical shifts (δ) are reported in parts per million (ppm) downfield from tetramethylsilane and for ^1^H NMR are referenced to residual proton in the NMR solvent (CDCl_3_ = δ 7.26 ppm). ^13^C NMR are referenced to the residual solvent peak (CDCl_3_ = δ 77.16 ppm). All ^13^C spectra are ^1^H decoupled. NMR data are represented as follows: chemical shift (δ ppm), integration, multiplicity (s = singlet, d = doublet, t = triplet, q = quartet, dd = double doublet, m = multiplet, app. d = apparent doublet, app. t. = apparent triplet), coupling constant (*J*) in Hertz (Hz). High resolution mass spectra [electrospray ionisation (ESI-TOF)] (HRMS) were measured on a micromass LCT orthogonal time-of-flight mass spectrometer with leucine enkephalin (Tyr-Gly-Phe-Leu) as an internal lock mass. Infrared spectra were recorded on a FT-IR spectrometer and are reported in terms of wavenumbers (ν_max_) with units of reciprocal centimetres (cm^−1^). Microwave experiments were conducted in a CEM Discover S-class microwave reactor with controlled irradiation at 2.45 GHz using standard microwave process Pyrex vials. Reaction time reflects time at the set reaction temperature maintained by cycling of irradiation (fixed hold times). Optical rotation (α) values were measured at room temperature and specific rotation ([α]_D_^20^) values are given in deg.dm^−1^.cm^3^.g^−1^. Melting points were determined in open capillary tubes. HPLC analysis was carried out on a Shimadzu LC-10AT _vp_ machine and Schimadzu LC-2010A machine equipped with a UV-Vis detector employing Chiracel^®^ OD (Sigma Aldrich) and AD columns from Diacel Chemical Industries (Illkirch, France).

### 3.1. 4-Chloro-Ferrocenylbutanone (***10***)

Ferrocene 9 (12.7 g, 68 mmol) was added to a dry 500-mL two-necked room-bottom flask (RBF) containing a magnetic stir bar under an inert atmosphere. Dry CH_2_Cl_2_ (120 mL) was added to the reaction flask which was cooled to 0 °C. 4-Chlorobutyryl chloride (95%, 7.3 mL, 62 mmol) was added to the reaction mixture followed by the slow addition of aluminium chloride (9.9 g, 74 mmol). The reaction mixture was warmed to room temperature and stirred for 18 h. Ice-cold H_2_O (100 mL) was added to reaction mixture followed by 10% Na_2_S_2_O_4_ solution (100 mL). The mixture was stirred for 30 min and the aqueous layer was extracted with CH_2_Cl_2_ (4 × 50 mL). The combined organic layers were washed with NaOH (2 M, 100 mL) and brine (100 mL), and dried with anhydrous Na_2_SO_4_. The solvent was removed in vacuo, and the crude product was purified by silica gel column chromatography (pentane/EtOAc) to yield 10 as an orange solid (17.3 g, 96%). Spectroscopic data are in good accordance to literature [14].

### 3.2. (R)-4-Chloro-1-Ferrocenylbutanol ((R)-***11***)

BH_3._THF (3 mL, 1.0 M, 3 mmol) was added to (*S*)-(−)-2-methyl-CBS-oxazaborolidine (crude residue) in a 250 mL Schlenk flask containing a magnetic stir bar under an inert atmosphere, and the reaction flask was cooled to −55 °C. A solution of 4-Chloro-ferrocenylbutanone (**11**) (3.67 g, 12.6 mmol) in dry THF (90 mL) was added followed by another portion of BH_3_.THF (1.0 M, 6 mL, 6 mmol). The reaction mixture was stirred for 18 h. The reaction mixture was warmed to 0 °C and then quenched by slow dropwise addition of MeOH (20 mL). The solvent was removed in vacuo, and the crude product was purified by silica gel column chromatography (pentane/EtOAc) to yield (*R*)-**11** as an orange oil (3.12 g, >99%, 94.5% *ee*). Spectroscopic data are in good accordance with the literature [14].

### 3.3. (R)-4-Chloro-2-Acetoxy-1-Ferrocenylbutane ((R)-***12***)

(*R*)-4-Chloro-1-ferrocenylbutanol ((*R)-***11**) (3.60 g, 12.3 mmol), 4-dimethylaminopyridine (0.075 mg, 0.62 mmol) and triethylamine (75 mL) were added to a dry 300-mL RBF containing a magnetic stir bar under an inert atmosphere. Acetic anhydride (1.76 mL, 18.6 mmol) was added to the reaction flask which was stirred at room temperature for 18 h. Et_2_O (100 mL) was added to reaction mixture which was subsequently washed with H_2_O (100 mL), 10% aqueous NH_4_Cl (2 × 40 mL), H_2_O (50 mL) and dried with anhydrous Na_2_SO_4_. The reaction mixture was filtered, and the solvent was removed in vacuo. The resultant crude orange oil was used directly in the next step without further purification. 

### 3.4. (R)-N-Allyl-Pyrrolidin-2′-ylferrocene ((R)-***13***)

(R)-4-Chloro-2-acetoxy-1-ferrocenylbutane (4.20 g, 12.5 mmol), allylamine (6.0 mL, 80.0 mmol) and dry MeOH (6.0 mL) were added to a dry sealed microwave vial containing a magnetic stir bar under an inert atmosphere. The reaction mixture was heated to reflux, stirred for 4 h then diluted with Et_2_O (10 mL), washed with sat. aqueous NaHCO_3_ (2 × 10 mL), brine (10 mL) and dried with anhydrous Na_2_SO_4_. The solvent was removed in vacuo and the crude product was purified by alumina column chromatography (pentane/EtOAc, 30:1 with 1% triethylamine) to yield (R)-**13** as an orange oil (3.61 g, 97%, 92.2% ee). Spectroscopic data are in good accordance with the literature [14].

### 3.5. 2-[(2R)-N-Allyl-Pyrrolidin-2′-yl]-(1S)-Diphenylphosphineferrocene (***14***) and 2-[(2R)-N-Allyl-Pyrrolidin-2′-yl]-(1R)-Diphenylphosphineferrocene (***14***)

(*R*)-N-allyl-pyrrolidin-2′-ylferrocene ((R)-**13**) (4.20 g, 12.5 mmol) and dry Et_2_O (6.0 mL) were added to a dry 250-mL RBF containing a magnetic stir bar under an inert atmosphere. The reaction mixture was cooled to −78 °C and *s*-BuLi (6.0 mL, 80.0 mmol) was added dropwise. After stirring for 3 h, the reaction mixture was warmed to 0 °C and stirred for an additional 1 h. Ph_2_PCl (6.0 mL, 80.0 mmol) was added, and the reaction mixture was stirred for 1.5 h and then quenched with aqueous NH_4_Cl (10%, 15 mL). The aqueous layer was separated and washed with CH_2_Cl_2_ (2 × 50 mL) brine (10 mL) and dried with anhydrous Na_2_SO_4_. The solvent was removed in vacuo and the crude product was purified by purified by silica gel column chromatography (pentane/EtOAc, 10:1 to 2:1) then alumina column chromatography (pentane/EtOAc with 0.1% triethylamine, 40:1 to 15:1) to yield (*R*,*S_p_*)-**14** as an orange solid (0.79 g, 24%, 93% *d.e.*) and (*R*,*R_p_*)-**14** as an orange solid (1.08 g, 33%, >99% *d.e.*). (*R*,*S_p_*)-**14** (0.56 g) was dissolved in 4.5 mL pentane then cooled to −20 °C for 30 min. The precipitate was filtered, washed with pentane, dried and collected to yield (*R*,*S_p_*)-**14** as an orange solid (0.43 g, >99% *d.e.*).

#### 3.5.1. Spectroscopic Analysis of (R,S_p_)-**14**

R_f_ = 0.22 (pentane/EtOAc 10:1); m.p. = 136–138 °C; [α]_D_^20^ = −232.8 (*c* 0.63, CH_2_Cl_2_); IR (neat): ν_max_ = 3054, 994 (C=C-H), 2939, 2922, 1443 (sp^3^C-H), 1628 (Alkene: C=C), 1609, 1587, 1565 (Aromatic: C=C) cm^−1^; ^1^H NMR (300 MHz, CDCl_3_): *δ* 7.66–7.57 (m, 2H), 7.41–7.35 (m, 3H), 7.35–7.18 (m, 5H), 5.67–5.48 (m, 1H), 4.82 (d, J = 10.1 Hz, 1H), 4.74 (d, J = 17.1 Hz, 1H), 4.57 (s, 1H), 4.37 (t, J = 2.3 Hz, 1H), 3.96 (s, 5H), 3.46 (td, J = 8.0, 3.2 Hz, 1H), 3.12–2.99 (m, 1H), 2.91 (dd, J = 13.3, 5.2 Hz, 1H), 2.50–2.34 (m, 1H), 2.27–2.14 (m, 1H), 2.13–1.96 (m, 2H), 1.93–1.68 (m, 2H) ppm; ^13^C NMR (126 MHz, CDCl_3_): *δ* 139.7 (d, *J* = 8.6 Hz), 137.9 (d, *J* = 8.5 Hz), 135.4 (d, *J* = 21.9 Hz), 132.9 (d, *J* = 18.9 Hz), 129.2, 128.2, 128.1 (d, *J* = 1.6 Hz), 128.1 (d, *J* = 1.6 Hz), 116.2, 75.4 (d, *J* = 8.7 Hz), 70.8 (d, *J* = 4.5 Hz), 70.8, 69.9, 69.7 (d, *J* = 4.3 Hz), 69.6, 62.1 (d, *J* = 9.5 Hz), 57.2, 54.0, 35.2, 22.6 ppm; ^31^P NMR (202 MHz, CDCl_3_) *δ* 25.7 ppm; HRMS (ESI-TOF): calcd. for C_29_H_31_NPFe [M + H]^+^ 480.1544; found 480.1536. See Appendix A, pages 21–22 for ^1^H, ^13^C and ^31^P NMR spectra.

#### 3.5.2. Spectroscopic Analysis of (R,R_p_)-**14**

R_f_ = 0.34 (pentane/EtOAc 9:1); m.p. = 110–112 °C; [α]_D_^20^ = 92.7 (*c* 0.4, CH_2_Cl_2_); IR (neat): ν_max_ = 3048, 979 (C=C-H), 1611 (Alkene: C=C) cm^−1^; ^1^H NMR (300 MHz, CDCl_3_): *δ* 7.66–7.48 (m, 2H), 7.45–7.31 (m, 3H), 7.29–7.11 (m, 5H), 5.69–5.49 (m, 1H), 5.12 (d, *J* = 16.8 Hz, 1H), 4.99 (d, *J* = 9.9 Hz, 1H), 4.44 (s, 1H), 4.28 (s, 1H), 3.97 (s, 4H), 3.87 (s, 2H), 3.40 (t, *J* = 7.8 Hz, 1H), 3.04 (t, *J* = 7.1 Hz, 1H), 2.66 (dd, *J* = 12.7, 8.2 Hz, 1H), 2.17–1.47 (m, 6H) ppm; ^13^C NMR (126 MHz, CDCl_3_): *δ* 140.6 (d, *J* = 9.1 Hz), 138.5 (d, *J* = 9.2 Hz), 137.1, 135.4 (d, *J* = 21.7 Hz), 132.6 (d, *J* = 18.3 Hz), 129.1, 128.07 (d, *J* = 7.8 Hz), 127.9 (d, *J* = 6.2 Hz), 127.7, 115.9, 97.6 (d, *J* = 22.5 Hz), 73.3 (d, *J* = 11.8 Hz), 71.7 (d, *J* = 4.7 Hz), 71.0 (d, *J* = 5.1 Hz), 69.7, 68.6, 63.5 (d, *J* = 3.9 Hz), 58.2, 54.6, 35.3 (d, *J* = 7.6 Hz), 22.8 (d, *J* = 1.2 Hz) ppm; ^31^P NMR (202 MHz, CDCl_3_) *δ* 22.98 ppm; HRMS (ESI-TOF): calcd. for C_29_H_31_NPFe [M + H]^+^ 480.1544; found 480.1563. See Appendix A, pages 23–24 for ^1^H, ^13^C and ^31^P NMR spectra.

### 3.6. 2-[(2R)-N-H-Pyrrolidin-2′-yl]-(1S)-Diphenylphosphineferrocene ((R,S_p_)-***15***)

2-[(2*R*)-N-allyl-pyrrolidin-2′-yl]-(1*S*)-diphenylphosphineferrocene (*R*,*S_p_*)-**14**) (0.41 g, 0.85 mmol), Pd(PPh_3_)_4_ (18.8 mg, 80.0 mmol), 1,3-dimethylbarbituric acid (NDMBA) (0.42 g, 2.69 mmol) and dry CH_2_Cl_2_ (9.0 mL) were added to a dry 50-mL Schlenk flask containing a magnetic stir bar under an inert atmosphere. The reaction mixture was heated to 35 °C, stirred for 5 h, and then quenched with sat. aqueous NaHCO_3_ (10 mL). The organic layer was separated and washed with sat. aqueous NaHCO_3_ (10 mL) and dried with anhydrous Na_2_SO_4_. The solvent was removed in vacuo and the crude product was purified by alumina column chromatography (pentane/EtOAc/MeOH/ triethylamine, 3:1:0.1:0.01) to yield (*R*,*S_p_*)-**15**) as a yellow solid (0.37 g, 98%).

#### Spectroscopic Analysis of (R,S_p_)-**15**)

R_f_ = 0.37 (pentane/EtOAc/MeOH, 3:1:0.1)**;** m.p. = 131–132 °C; [α]_D_^20^ = −184.4 (*c* 0.8, CH_2_Cl_2_); IR (neat): ν_max_ = 3049, 990 (C=C-H), 2989, 1432 (sp^3^C-H), 1662 (Alkene: C=C) cm^−1^; ^1^H NMR (300 MHz, CDCl_3_): *δ* 7.59–7.46 (m, 2H), 7.42–7.33 (m, 3H), 7.31–7.15 (m, 5H), 4.46 (s, 1H), 4.34–4.20 (m, 2H), 4.04 (s, 5H), 3.74–3.66 (m, 1H), 2.93 (dd, *J* = 14.1, 7.7 Hz, 1H), 2.76 (dd, *J* = 15.3, 8.2 Hz, 1H), 2.27–2.09 (m, 1H), 2.03–1.70 (m, 3H) ppm; ^13^C NMR (126 MHz, CDCl_3_): *δ* 140.2 (d, *J* = 10.0 Hz), 137.4 (d, *J* = 8.9 Hz), 135.4 (d, *J* = 20.9 Hz), 132.7 (d, *J* = 18.4 Hz), 129.2, 128.4 (d, *J* = 6.1 Hz), 128.3, 128.2, 96.04 (d, *J* = 22.8 Hz), 75.8 (d, *J* = 6.5 Hz), 71.4 (d, *J* = 4.0 Hz), 69.6, 69.5, 69.1 (d, *J* = 3.9 Hz), 56.4, 56.3, 47.0, 31.6, 25.3 ppm; ^31^P NMR (202 MHz, CDCl_3_) *δ* 23.6 ppm; HRMS (ESI-TOF): calcd. for C_26_H_27_NPFe [M + H]^+^ 440.1231; found 440.1243. See Appendix A, pages 25−26 for ^1^H, ^13^C and ^31^P NMR spectra.

### 3.7. 2-[(2R)-N-H-Pyrrolidin-2′-yl]-(1R)-Diphenylphosphineferrocene)-(R,R_p_)-***15***)

Prepared according to the same procedure as for (*R,S_p_*)-**15** using (*R*,*R_p_*)-**14** (1.80 g, 3.75 mmol), to afford the product as a yellow solid (1.42 g, 86%).

#### 3.7.1. Spectroscopic Analysis of (R,R_p_)-**15**

R_f_ = 0.37 (pentane/EtOAc/MeOH, 3:1:0.1)**;** m.p. = 139–140 °C; [α]_D_^20^ = 239.0 (*c* 0.75, CH_2_Cl_2_); IR (neat): ν_max_ = 3054, 997 (C=C-H), 2987, 1444 (sp^3^C-H), 1636, 1590 (Aromatic: C=C) cm^−1^; ^1^H NMR (300 MHz, CDCl_3_): *δ* 7.61–7.44 (m, 2H), 7.41–7.33 (m, 3H), 7.30–7.13 (m, 5H), 4.50 (s, 1H), 4.30–4.16 (m, 2H), 4.07 (s, 5H), 3.77–3.66 (m, 1H), 3.13 (dd, *J* = 13.3, 8.4 Hz, 1H), 2.93 (dd, *J* = 16.4, 7.7 Hz, 1H), 2.18 (s, 1H), 1.84–1.45 (m, 3H), 1.21–1.02 (m, 1H) ppm; ^13^C NMR (126 MHz, CDCl_3_): *δ* 140.3 (d, *J* = 9.9 Hz), 137.4 (d, *J* = 9.0 Hz), 135.1 (d, *J* = 20.9 Hz), 132.7 (d, *J* = 18.5 Hz), 129.1, 128.2, 128.2, 128.1 (d, *J* = 2.9 Hz), 98.4 (d, *J* = 22.5 Hz), 74.8 (d, *J* = 7.6 Hz), 71.1 (d, *J* = 4.4 Hz), 69.5, 69.0, 67.8 (d, *J* = 4.1 Hz), 56.7 (d, *J* = 11.3 Hz), 46.8, 34.9, 25.9 ppm; ^31^P NMR (202 MHz, CDCl_3_) *δ* 23.9 ppm; HRMS (ESI-TOF): calcd. for C_26_H_27_NPFe [M + H]^+^ 440.1231; found 440.1209. See Appendix A, pages 27–28 for ^1^H, ^13^C and ^31^P NMR spectra.

#### 3.7.2. Typical Procedure A: Phosphine-Coupling

(*R*,*Sp*)-**15** or (*R*,*Rp*)-**15** (1.0 equiv.), Et_3_N (3.0 equiv.) and dry toluene (0.23 M) were added to a dry 25-mL Schlenk flask containing a magnetic stir bar under an inert atmosphere. The di-substituted chlorophosphine (1.0 equiv.) in dry toluene (0.23 M) was added, and the reaction mixture was stirred at room temperature for 18 h. Heptane (5 mL) was added, and the reaction mixture was filtered. The solvent was removed in vacuo and the crude product was purified by alumina column chromatography (pentane/EtOAc, 99:1 to pentane/EtOAc/MeOH, 3:1:0.1) to yield the product.

### 3.8. 2-[(2R)-N-Diphenylphosphine-Pyrrolidin-2′-yl]-(1S)-Diphenylphosphineferrocene (***L1***)

Prepared according to typical procedure A using chlorodiphenylphosphine 0.075 g, 0.340 mmol) to afford the product as an orange solid (0.165 g, 78%).

#### Spectroscopic Analysis of **L1**

R_f_ = 0.80 (pentane/EtOAc, 20:1); m.p. = 58–60 °C; [α]_D_^20^ = −146.0 (*c* 0.7, CH_2_Cl_2_); IR (neat): ν_max_ = 3032 (C=C-H), 2969, 1422 (sp^3^C-H) cm^−1^; ^1^H NMR (300 MHz, CDCl_3_): *δ* 7.67–7.58 (m, 2H), 7.40–7.34 (m, 3H), 7.29–7.13 (m, 10H), 7.11–6.99 (m, 5H), 4.78–4.66 (m, *J* = 7.1, 3.4 Hz, 1H), 4.49 (d, *J* = 1.1 Hz, 1H), 4.36 (t, *J* = 2.4 Hz, 1H), 3.92 (s, 5H), 3.09 (dtd, *J* = 10.0, 7.3, 2.6 Hz, 1H), 2.78–2.63 (m, 1H), 2.45–2.16 (m, 2H), 1.90–1.70 (m, 1H), 1.55–1.34 (m, 1H) ppm; ^13^C NMR (126 MHz, CDCl_3_): *δ* 140.0 (d, *J* = 7.8 Hz, 2C), 139.7 (d, *J* = 22.8 Hz), 139.1 (d, *J* = 9.1 Hz), 135.6 (d, *J* = 22.2 Hz, 2C), 133.0 (d, *J* = 21.0 Hz, 2C), 132.8 (d, *J* = 17.7 Hz), 132.8 (d, *J* = 17.7 Hz), 131.4 (d, *J* = 18.5 Hz, 2C), 129.1, 128.4, 128.1 (d, *J* = 8.1 Hz, 2C), 127.9 (d, *J* = 6.1 Hz, 2C), 127.8 (d, *J* = 5.3 Hz, 2C), 127.7 (d, *J* = 6.0 Hz, 2C), 127.4, 127.4, 99.3 (dd, *J* = 24.7, 5.4 Hz), 75.8 (d, *J* = 10.0 Hz), 71.4 (d, *J* = 4.4 Hz), 69.8 (dd, *J* = 4.4, 2.5 Hz), 69.7, 69.6 (5C), 60.9 (dd, *J* = 31.1, 10.0 Hz), 47.7 (d, *J* = 10.0 Hz), 36.6 (d, *J* = 6.8 Hz), 26.0 ppm; ^31^P NMR (202 MHz, CDCl_3_) *δ* 44.8 (d, *J* = 16.4 Hz), -24.3 (d, *J* = 16.4 Hz) ppm; HRMS (ESI-TOF): calcd. for C_38_H_36_NP_2_Fe [M + H]^+^ 624.1672; found 624.1650. See Appendix A, pages 29–31 for ^1^H, ^13^C and ^31^P and ^31^P-^31^P COSY NMR spectra.

### 3.9. 2-[(2R)-N-Bis(2-Methylphenyl)phosphine-Pyrrolidin-2′-yl]-(1S)-Diphenylphosphineferrocene (***L2***)

Prepared according to typical procedure A using bis(2-methylphenyl)chlorophosphine (0.057 g, 0.230 mmol) to afford the product as an orange solid (0.063 g, 42%).

#### Spectroscopic Analysis of **L2**

R_f_ = 0.79 (pentane/EtOAc 10:1); m.p. = 84–87 °C; [α]_D_^20^ = −220.7 (c 0.18, CH_2_Cl_2_); IR (neat): ν_max_ = 3047, 923 (C=C-H), 2921, 2858, 1444 (sp^3^C-H), 1586, 1563, 1502 (Aromatic: C=C) cm^−1^; ^1^H NMR (400 MHz, CDCl_3_): δ 7.65–7.54 (m, 2H), 7.40–7.28 (m, 4H), 7.23–6.89 (m, 11H), 6.74 (dd, *J* = 7.3, 2.6 Hz, 1H), 4.95–4.84 (m, 1H), 4.47 (d, *J* = 1.1 Hz, 1H), 4.31 (t, *J* = 2.4 Hz, 1H), 4.03–3.95 (m, 1H), 3.86 (s, 5H), 3.12 (dd, *J* = 16.7, 6.8 Hz, 1H), 2.68–2.31 (m, 3H), 2.25 (s, 3H), 1.99–1.83 (m, 1H), 1.69 (s, 3H), 1.65–1.45 (m, 1H) ppm; ^13^C NMR (126 MHz, CDCl_3_): δ 142.0 (d, *J* = 27.9 Hz), 140.5 (dd, *J* = 7.3, 1.3 Hz), 140.2 (d, *J* = 25.5 Hz), 139.8 (dd, *J* = 9.7, 2.1 Hz), 139.0 (d, *J* = 11.5 Hz), 136.9 (d, *J* = 18.2 Hz), 135.6 (d, *J* = 22.6 Hz, 2C), 132.4 (d, *J* = 17.2 Hz), 132.4 (d, *J* = 17.2 Hz), 131.4 (d, *J* = 1.1 Hz), 131.0 (d, *J* = 3.2 Hz), 129.8 (d, *J* = 24.6 Hz), 129.8 (d, *J* = 24.2 Hz), 128.9, 128.0 (d, *J* = 27.5 Hz, 2C), 127.8, 127.3 (d, *J* = 12.5 Hz, 2C), 127.3, 126.8, 125.1 (d, *J* = 15.2 Hz, 2C), 100.1 (dd, *J* = 25.5, 2.7 Hz), 75.8 (d, *J* = 10.7 Hz), 71.2 (d, *J* = 4.6 Hz), 69.8, 69.4 (5C), 69.0 (d, *J* = 4.3 Hz), 60.3 (dd, *J* = 32.9, 10.4 Hz), 48.7 (d, *J* = 9.9 Hz), 36.5 (dd, *J* = 6.5, 1.9 Hz), 26.9, 21.3 (dd, *J* = 20.8, 1.4 Hz), 20.7 (d, *J* = 18.8 Hz) ppm; ^31^P NMR (202 MHz, CDCl_3_) δ 31.6 (d, *J* = 26.0 Hz), −24.8 (d, *J* = 26.0 Hz) ppm; HRMS (ESI-TOF): calcd. for C_40_H_40_NP_2_Fe [M + H]^+^ 652.1985; found 652.1993. See Appendix A, pages 32–33 for ^1^H, ^13^C and ^31^P NMR spectra.

### 3.10. 2-[(2R)-N-Bis(4-fluorophenyl)phosphine-Pyrrolidin-2′-yl]-(1S)-Diphenylphosphineferrocene (***L3***)

Prepared according to typical procedure A using bis(4-fluorophenyl)chlorophosphine (0.054 g, 0.210 mmol) to afford the product as a yellow solid (0.072 g, 50%).

#### Spectroscopic Analysis of **L3**

R_f_ = 0.655 (pentane/EtOAc, 20:1); m.p. = 72–74 °C; [α]_D_^20^ = –166.0 (*c* 0.2, CHCl_3_); IR (neat): ν_max_ = 3022, 982 (C=C-H), 2936, 2911, 1486 (sp^3^C-H), 1647 (Alkene: C=C) cm^−1^; ^1^H NMR (300 MHz, CDCl_3_): *δ* 7.67–7.57 (m, 2H), 7.41–7.34 (m, 3H), 7.26–7.18 (m, 2H), 7.10–6.82 (m, 11H), 4.83–4.68 (m, 1H), 4.43 (s, 1H), 4.38 (t, *J* = 2.4 Hz, 1H), 4.03 (d, *J* = 1.0 Hz, 1H), 3.91 (s, 5H), 3.10–2.96 (m, 1H), 2.76–2.64 (m, 1H), 2.41–2.15 (m, 2H), 1.88–1.72 (m, 1H), 1.55–1.37 (m, 1H) ppm; ^13^C NMR (126 MHz, CDCl_3_): *δ* 164.1, 163.5, 162.1, 161.5, 139.9–139.7 (m), 138.7 (dd, *J* = 8.8, 1.3 Hz), 135.4 (d, *J* = 22.2 Hz, 2C), 134.6 (d, *J* = 22.7 Hz), 134.6 (d, *J* = 22.5 Hz), 133.0 (d, *J* = 19.8 Hz), 132.9 (d, *J* = 19.8 Hz), 132.7 (d, *J* = 17.8 Hz), 132.7 (d, *J* = 17.7 Hz), 129.0, 128.0 (d, *J* = 8.2 Hz, 2C), 127.5 (d, *J* = 6.1 Hz, 2C), 127.3, 115.0 (d, *J* = 6.7 Hz), 114.9 (d, *J* = 5.9 Hz), 114.8 (d, *J* = 6.7 Hz), 114.7 (d, *J* = 5.8 Hz), 98.5 (dd, *J* = 25.2, 6.3 Hz), 75.8 (d, *J* = 10.8 Hz), 71.38 (d, *J* = 4.6 Hz), 69.67, 69.47 (5C), 69.34 (dd, *J* = 4.4, 2.2 Hz), 60.85 (dd, *J* = 31.6, 10.6 Hz), 47.2 (d, *J* = 10.0 Hz), 35.9 (d, *J* = 3.9 Hz), 25.7 ppm; ^19^F NMR (470 MHz, CDCl_3_) δ −113.1–−113.2 (m), −114.7–−114.8 (m) ppm; ^31^P NMR (202 MHz, CDCl_3_) δ 43.1 (dt, *J* = 19.5, 5.4 Hz), −25.1 (d, *J* = 19.5 Hz) ppm; HRMS (ESI-TOF): calcd. for C_38_H_33_NP_2_ F_2_Fe [M + H]^+^ 660.1484; found 660.1481. See Appendix A, pages 34–36 for ^1^H, ^13^C, ^31^P and ^19^F NMR spectra.

### 3.11. 2-[(2R)-N-Bis(3,5-Di-Trifluoromethylphenyl)phosphine-Pyrrolidin-2′-yl]-(1S)-Diphenylphosphineferrocene (***L4***)

Prepared according to typical procedure A using bis(3,5-di-trifluoromethylphenyl)chlorophosphine (0.103 g, 0.210 mmol) to afford the product as an orange solid (0.075 g, 40%).

#### Spectroscopic Analysis of **L4**

R_f_ = 0.37 (pentane/EtOAc/MeOH, 3:1:0.1); m.p. = 56–58 °C; [α]_D_^20^ = −184.3 (*c* 0.08, CH_2_Cl_2_); IR (neat): ν_max_ = 3053, 987 (C=C-H), 2970, 1434 (sp^3^C-H), 1587, 1576 (Aromatic: C=C) cm^−1^; ^1^H NMR (300 MHz, CDCl_3_): *δ* 7.76 (s, 2H), 7.68–7.57 (m, 2H), 7.44–7.33 (m, 7H), 7.27–7.17 (m, 2H), 6.93–6.83 (m, 3H), 5.07–4.94 (m, 1H), 4.51 (t, *J* = 2.4 Hz, 1H), 4.43 (s, 1H), 4.21 (t, *J* = 2.9 Hz, 1H), 3.91 (s, 5H), 2.89 (dd, *J* = 16.4, 8.3 Hz, 1H), 2.74–2.60 (m, 1H), 2.57–2.41 (m, 1H), 2.34–2.15 (m, 1H), 2.02–1.85 (m, 1H), 1.82–1.65 (m, 1H) ppm; ^13^C NMR (126 MHz, CDCl_3_): *δ* 142.1 (d, *J* = 1.0 Hz), 141.9 (d, *J* = 1.3 Hz), 141.6, 141.4, 138.9 (dd, *J* = 6.1, 1.8 Hz), 138.3 (dd, *J* = 7.8, 1.7 Hz), 135.3 (d, *J* = 22.4Hz, 2C), 132.9 (d, *J* = 18.6 Hz), 132.8 (d, *J* = 18.6 Hz), 131.8–131.7 (m), 131.7–131.5 (m), 131.4 (d, *J* = 4.7 Hz), 131.3–131.1 (m, 2C), 131.0–130.9 (m, *J* = 23.5 Hz), 129.1, 128.1 (d, *J* = 8.4 Hz, 2C), 127.6, 127.3 (d, *J* = 6.7 Hz, 2C), 122.6–122.4 (m), 122.2–121.9 (m), 96.6 (dd, *J* = 26.5, 5.9 Hz), 76.5 (dd, *J* = 10.0, 1.4 Hz), 72.0 (d, *J* = 4.7 Hz), 70.6, 69.7 (5C), 68.4 (d, *J* = 3.5 Hz), 60.9 (dd, *J* = 32.6, 12.7 Hz), 47.3 (d, *J* = 10.8 Hz), 35.3 (d, *J* = 5.5 Hz), 26.0 ppm; ^19^F NMR (470 MHz, CDCl_3_) δ -62.3, −61.0 ppm; ^31^P NMR (202 MHz, CDCl_3_) *δ* 40.1 (d, *J* = 26.2 Hz), −27.1 (d, *J* = 26.2 Hz) ppm; HRMS (ESI-TOF): calcd. for C_42_H_32_NP_2_ F_12_Fe [M + H]^+^ 896.1168; found 896.1155. See Appendix A, pages 37–39 for ^1^H, ^13^C, ^31^P and ^19^F NMR spectra.

### 3.12. 2-[(2R)-N-(R)-1,1′-Binaphthyl-2,2′-Diylphosphoro-Pyrrolidin-2′-yl]-(1S)-Diphenylphosphineferrocene (***L5***)

Prepared according to typical procedure A using (*R*)-1,1′-binaphthyl-2,2′-diyl phosphorochloridate (0.21 g, 0.60 mmol, 1.3 equiv.) to afford the product as a yellow solid (0.29 g, 83%).

#### Spectroscopic Analysis of **L5**

R_f_ = 0.47 (pentane/EtOAc, 9:1); m.p. = 219–220 °C; [α]_D_^20^ = −241.2 (*c* 0.29, CH_2_Cl_2_); IR (neat): ν_max_ = 3054, 896 (C=C-H), 2987, 1421 (sp^3^C-H) cm^−1^; ^1^H NMR (300 MHz, CDCl_3_): *δ* 7.93–7.78 (m, 4H), 7.67–7.57 (m, 2H), 7.47–7.27 (m, 10H), 7.25–7.13 (m, 5H), 6.92 (d, *J* = 8.8 Hz, 1H), 5.24–5.11 (m, 1H), 4.45 (s, 1H), 4.35 (d, *J* = 2.3 Hz, 1H), 3.96 (s, 5H), 3.90 (s, 1H), 2.86–2.70 (m, 1H), 2.66–2.54 (m, 1H), 2.51–2.27 (m, 2H), 1.87–1.60 (m, 2H) ppm; ^13^C NMR (126 MHz, CDCl_3_): *δ* 150.4, 149.9, 140.4 (d, *J* = 8.9 Hz), 138.5 (d, *J* = 6.4 Hz), 135.6 (d, *J* = 22.1 Hz, 2C), 132.7, 132.6 (d, *J* = 1.6 Hz), 132.5 (d, *J* = 1.9 Hz), 132.4 (d, *J* = 2.0 Hz), 131.20, 130.4, 129.8, 129.4, 129.1, 128.4, 128.2, 128.1, 128.1, 128.0, 128.0, 127.9, 127.4, 127.3, 127.0 (2C), 125.8 (d, *J* = 10.9 Hz, 2C), 124.5, 124.3, 124.3, 123.91–123.81 (m), 122.3 (d, *J* = 1.8 Hz), 122.2 (d, *J* = 1.6 Hz), 121.9, 97.1 (dd, *J* = 24.4, 7.4 Hz), 75.5 (d, *J* = 11.2 Hz), 71.5 (d, *J* = 4.4 Hz), 69.7–69.6 (m, 6C), 69.5, 56.1 (dd, *J* = 33.6, 8.8 Hz), 44.0 (d, *J* = 6.8 Hz), 34.0, 25.34 ppm; ^31^P NMR (202 MHz, CDCl_3_) *δ* 146.9 (d, *J* = 56.2 Hz), −23.8 (d, *J* = 56.2 Hz) ppm; HRMS (ESI-TOF): calcd. for C_46_H_38_NO_2_P_2_Fe [M + H]^+^ 754.1727; found 754.1719. See Appendix A, pages 40–41 for ^1^H, ^13^C, and ^31^P NMR spectra and page 40 for X-ray crystallographic data.

### 3.13. 2-[(2R)-N-(S)-1,1′-Binaphthyl-2,2′-Diylphosphoro-Pyrrolidin-2′-yl]-(1S)-Diphenylphosphineferrocene (***L6***)

Prepared according to typical procedure A using (*S*)-1,1′-binaphthyl-2,2′-diyl phosphorochloridate (0.10 g, 0.29 mmol, 1.2 equiv.) to afford the product as a yellow solid (0.12 g, 70%).

#### Spectroscopic Analysis of **L6**

R_f_ = 0.83 (pentane/EtOAc, 4:1); m.p. = 208–209 °C; [α]_D_^20^ = −32.5 (*c* 0.17, CH_2_Cl_2_); IR (neat): ν_max_ = 2939, 2926, 1443 (sp^3^C-H), 1613, 1588, 1503 (Aromatic: C=C) cm^−1^; ^1^H NMR (300 MHz, CDCl_3_): *δ* 7.92–7.81 (m, 4H), 7.65–7.55 (m, 2H), 7.43–7.33 (m, 6H), 7.30–7.19 (m, 9H), 7.15 (d, *J* = 8.7 Hz, 1H), 5.07–4.93 (m, 1H), 4.60 (s, 1H), 4.42 (t, *J* = 2.4 Hz, 1H), 4.01–3.97 (m, 1H), 3.95 (s, 5H), 3.03–2.91 (m, 1H), 2.56–2.42 (m, 1H), 2.39–2.23 (m, 2H), 1.82–1.67 (m, 1H), 1.64–1.49 (m, 1H) ppm; ^13^C NMR (126 MHz, CDCl_3_): *δ* 150.1 (d, *J* = 6.2 Hz), 149.8 (d, *J* = 1.8 Hz), 139.6 (d, *J* = 8.2 Hz), 138.5 (d, *J* = 9.1 Hz), 135.4 (d, *J* = 22.1 Hz, 2C), 132.8 (d, *J* = 4.4 Hz), 132.8, 132.6 (d, *J* = 2.8 Hz), 132.6 (d, *J* = 1.0 Hz), 131.1, 130.9, 130.6, 129.8, 129.4, 129.1, 128.8, 128.2, 128.1, 128.1, 128.0, 127.9, 127.9, 127.6, 126.9 (d, *J* = 9.1 Hz, 2C), 125.8 (d, *J* = 8.2 Hz, 2C), 124.4 (d, *J* = 7.6 Hz, 2C), 123.8, 122.9, 122.2, 122.2 (d, *J* = 1.5 Hz), 98.0 (dd, *J* = 24.2, 7.7 Hz), 75.6 (d, *J* = 11.2 Hz), 71.6 (d, *J* = 4.4 Hz), 69.7 (dd, *J* = 4.0, 2.2 Hz), 69.6 (5C), 68.2, 57.3 (dd, *J* = 35.3, 9.7 Hz), 44.1 (d, *J* = 4.8 Hz), 34.7, 26.0 ppm; ^31^P NMR (202 MHz, CDCl_3_) *δ* 150.8 (d, *J* = 29.3 Hz), -24.7 (d, *J* = 29.3 Hz) ppm; HRMS (ESI-TOF): calcd. for C_46_H_38_NO_2_P_2_Fe [M + H]^+^ 754.1727; found 754.1757. See Appendix A, pages 42–43 for ^1^H, ^13^C, and ^31^P NMR spectra.

### 3.14. 2-[(2R)-N-(R)-1,1′-Binaphthyl-2,2′-Diylphosphoro-Pyrrolidin-2′-yl]-(1R)-Diphenylphosphineferrocene (***L7***)

Prepared according to typical procedure A using (*R*)-1,1′-binaphthyl-2,2′-diyl phosphorochloridate (0.15 g, 0.44 mmol, 1.3 equiv.) to afford the product as an orange solid (0.23 g, 90%). 

#### Spectroscopic Analysis of **L7**

R_f_ = 0.86 (pentane/EtOAc 9:1); m.p. = 177–179 °C; [*a*]_D_^20^ = −83.5 (*c* 0.18, CH_2_Cl_2_); IR (neat): ν_max_ = 3052, 947 (C=C-H), 2967, 1434 (sp^3^C-H), 1588, 1567 (Aromatic: C=C) cm^−1^; ^1^H NMR (300 MHz, CDCl_3_): *δ* 8.04–7.83 (m, 5H), 7.68–7.55 (m, 3H), 7.47–7.35 (m, 8H), 7.33 (s, 1H), 7.31–7.26 (m, 3H), 7.25–7.19 (m, 2H), 5.32 (dd, *J* = 11.7, 7.4 Hz, 1H), 4.51 (d, *J* = 1.4 Hz, 1H), 4.31 (dd, *J* = 4.1, 1.7 Hz, 1H), 4.15 (s, 5H), 3.98–3.92 (m, 1H), 3.07–2.90 (m, 1H), 2.79–2.60 (m, 1H), 2.00–1.81 (m, 1H), 1.56–1.32 (m, 3H) ppm; ^13^C NMR (126 MHz, CDCl_3_): *δ* 151.0 (d, *J* = 4.7 Hz), 150.0, 140.3 (d, *J* = 9.1 Hz), 137.6 (d, *J* = 8.0 Hz), 135.3 (d, *J* = 21.6 Hz, 2C), 132.8 (d, *J* = 15.6 Hz), 132.5 (d, *J* = 18.1 Hz, 2C), 131.4, 130.5, 130.3, 129.7, 129.2, 128.3, 128.2, 128.1, 128.1, 128.0, 128.0, 127.9, 127.0 (d, *J* = 10.7 Hz, 2C), 126.1 (d, *J* = 4.1 Hz, 2C), 124.8, 124.4, 124.2 (d, *J* = 5.1 Hz), 122.1 (d, *J* = 2.0 Hz), 122.0 (2C), 99.9 (d, *J* = 28.1 Hz), 71.9 (d, *J* = 10.4 Hz), 71.5 (d, *J* = 4.1 Hz), 70.2 (d, *J* = 4.7 Hz), 70.0 (d, *J* = 5.7 Hz, 5C), 68.5, 57.4 (dd, *J* = 34.8, 8.6 Hz), 45.3 (d, *J* = 6.8 Hz), 36.1 (t, *J* = 3.7 Hz), 23.91 ppm; ^31^P NMR (202 MHz, CDCl_3_) *δ* 148.1, −24.4 ppm; HRMS (ESI-TOF): calcd. for C_46_H_38_NO_2_P_2_Fe [M + H]^+^ 754.1727; found 754.1719. See Appendix A, pages 44–45 for ^1^H, ^13^C, and ^31^P NMR spectra.

### 3.15. 2-[(2R)-N-(S)-1,1′-Binaphthyl-2,2′-Diylphosphoro-Pyrrolidin-2′-yl]-(1R)-Diphenylphosphineferrocene (***L8***)

Prepared according to typical procedure A using (*S*)-1,1′-binaphthyl-2,2′-diyl phosphorochloridate (0.15 g, 0.44 mmol, 1.3 equiv.) to afford the product as an orange solid (0.25 g, 98%).

#### 3.15.1. Spectroscopic Analysis of **L8**

R_f_ = 0.91 (pentane/EtOAc 9:1); m.p. = 185–187 °C; [a]_D_^20^ = 380.2 (*c* 1.3, CHCl_3_); IR (neat): ν_max_ = 3067, 977 (C=C-H), 1619 (Alkene: C=C) cm^−1^; ^1^H NMR (300 MHz, CDCl_3_): *δ* 8.04–7.90 (m, 5H), 7.70–7.52 (m, 4H), 7.50–7.34 (m, 8H), 7.33–7.26 (m, 3H), 7.25–7.22 (m, 2H), 5.33 (dd, *J* = 9.6, 7.9 Hz, 1H), 4.68 (d, *J* = 1.3 Hz, 1H), 4.41 (t, *J* = 2.2 Hz, 1H), 4.18 (s, 5H), 4.01–3.95 (m, 1H), 3.27–3.11 (m, 1H), 2.53–2.40 (m, 1H), 1.79–1.61 (m, 1H), 1.55–1.34 (m, 2H), 1.23–1.12 (m, 1H) ppm; ^13^C NMR (126 MHz, CDCl_3_): *δ* 150.1 (d, *J* = 6.5 Hz), 149.9 (d, *J* = 2.1 Hz), 140.3 (d, *J* = 9.2 Hz), 137.61 (d, *J* = 8.1 Hz), 135.2 (d, *J* = 21.5 Hz, 2C), 132.9 (d, *J* = 1.3 Hz), 132.7 (d, *J* = 1.1 Hz), 132.5 (d, *J* = 18.3 Hz, 2C), 131.3, 130.7, 130.3, 129.9, 129.2, 128.4, 128.2, 128.0, 128.1, 128.0, 128.0, 127.9, 127.0 (d, *J* = 21.6 Hz, 2C), 126.1 (d, *J* = 4.9 Hz, 2C), 124.6 (d, *J* = 5.2 Hz, 2C), 123.8 (d, *J* = 4.9 Hz), 123.3 (d, *J* = 2.3 Hz), 122.0, 100.9 (dd, *J* = 24.9, 1.9 Hz), 71.8 (d, *J* = 10.4 Hz), 71.4 (d, *J* = 4.2 Hz), 69.9 (d, *J* = 4.7 Hz, 5C), 69.7 (d, *J* = 4.9 Hz), 68.4, 59.5 (dd, *J* = 41.3, 9.3 Hz), 44.0 (d, *J* = 5.9 Hz), 36.0–35.3 (m), 24.6 ppm; ^31^P NMR (202 MHz, CDCl_3_) *δ* 150.8, −24.7 ppm; HRMS (ESI-TOF): calcd. for C_46_H_38_NO_2_P_2_Fe [M + H]^+^ 754.1727; found 754.1765. See Appendix A, pages 46–48 for ^1^H, ^13^C and ^31^P and ^31^P-^31^P COSY NMR spectra.

#### 3.15.2. Rhodium-Catalyzed Asymmetric Hydrogenation

##### Preparation of Substrates/Characterization Data for Substrates and Products

The substrates for catalysis were prepared according to the literature procedures and all characterization data for the substrates and products were in accordance with those reported. **16a**/**17a**–**16c**/**7c** and **16e**/**17e** and **16g**/**17g**, [27] **16d**/**16d** and **16i**/**17i** and **17k** and **17a**/**19a**-**18e**/**19e**, [37] **16k** is commercially available, **16f**/**17f**, [38] **16h**/**17h** and **16j**/**17j**, [39] and **16f**/**17f** [24].

#### 3.15.3. Rhodium-Catalyzed Asymmetric Hydrogenation of Dehydroamino Acid Esters

##### Typical Procedure B: Optimization and Substrate Scope

Optimization

The Rh source (0.005 mmol), ligand (0.006 mmol), substrate **16a**–**k** (0.5 mmol), and the solvent (2 mL) were added to a dry 10-mL Schlenk flask containing a magnetic stir bar under an inert atmosphere. The reaction mixture was cooled under liquid nitrogen, the atmosphere was evacuated (high vacuum), and then the reaction chamber was refilled with hydrogen using a balloon (reactions requiring higher pressures of hydrogen were quickly transferred to an autoclave). The reaction was stirred for the designated time, filtered through a plug of Celite^®^, and washed with the solvent of choice. The solvent was removed in vacuo to yield the crude product. The *ee* was determined by chiral HPLC and conversion of starting material to product by ^1^H NMR spectroscopy.

Substrate Scope

The reactions were performed with 0.5 mmol of the substrate using the procedure outlined above for the optimization process, with the following exceptions. Racemic reactions were performed with (±)-BINAP (1.1 mol %) using Rh(COD)_2_OTf (1.0 mol %), 2.3 bar H_2_ for substrate **16a** and 40 bar H_2_ for **16b**–**k**, in THF for 1-18h h at room temperature. Reactions with **L1** (1.1 mol %) were performed using Rh(COD)_2_OTf (1.0 mol %), 1 bar H_2_, in THF for 12 h at room temperature. Reactions with **L4** (1.1 mol %) were performed using Rh(COD)_2_OTf (1.0 mol %), 1 bar H_2_, in THF for 4 h at 0 °C. Reactions with **L7** (0.22 mol %) were performed using Rh(COD)_2_OTf (0.2 mol %), 10 bar H_2_, in THF for 12 h at room temperature. For the methods and chiral columns used to determine the enantiomeric excess, and chromatograms for racemic and enantioenriched products, see Appendix A and pages 4–14, respectively.

### 3.16. (S)-Methyl 2-Acetamido-3-Phenylpropanoate (***17a***)

Prepared according to typical procedure B to afford the product (> 99 % conversion, 99.5% ee with **L7**) with all characterization analysis in good accordance with the literature.

### 3.17. (S)-Methyl 2-Acetamido-3-(4-Methoxyphenyl)propanoate (***17b***)

Prepared according to typical procedure B to afford the product (>99% conversion, >99.9% ee with **L7**) with all characterization analysis in good accordance with the literature.

### 3.18. (S)-Methyl 2-Acetamido-3-(p-Tolyl)propanoate (***17c***)

Prepared according to typical procedure B to afford the product (>99 % conversion, 99.4 % ee with **L7**) with all characterization analysis in good accordance with the literature.

### 3.19. (S)-Methyl 2-Acetamido-3-(4-Chlorophenyl)propanoate (***17d***)

Prepared according to typical procedure B to afford the product (>99% conversion, 99.4% ee with **L7**) with all characterization analysis in good accordance with the literature.

### 3.20. (S)-Methyl 2-Acetamido-3-(4-Fluorophenyl)propanoate (***17e***)

Prepared according to typical procedure B to afford the product (>99% conversion, 98.5% ee with **L7**) with all characterization analysis in good accordance with the literature.

### 3.21. (S)-Methyl 2-Acetamido-3-(4-Nitrophenyl)propanoate (***17f***)

Prepared according to typical procedure B to afford the product (>99% conversion, 98.6% ee with **L7**) with all characterization analysis in good accordance with the literature.

### 3.22. (S)-Methyl 2-Acetamido-3-(3-Chlorophenyl)propanoate (***17g***)

Prepared according to typical procedure B to afford the product (>99% conversion, 99.4% ee with **L7**) with all characterization analysis in good accordance with the literature.

### 3.23. (S)-Methyl 2-Acetamido-3-(3-Bromophenyl)propanoate (***17h***)

Prepared according to typical procedure B to afford the product (>99% conversion, 99.0% ee with **L7**) with all characterization analysis in good accordance with the literature.

### 3.24. (S)-Methyl 2-Acetamido-3-(2-Chlorophenyl)propanoate (***17i***)

Prepared according to typical procedure B to afford the product (>99% conversion, 99.7% ee with **L7**) with all characterization analysis in good accordance with the literature.

### 3.25. (S)-Methyl 2-Acetamido-3-(Naphthalen-1-yl)propanoate (***17j***)

Prepared according to typical procedure B to afford the product (>99% conversion, 99.2% ee with **L7**) with all characterization analysis in good accordance with the literature.

### 3.26. (S)-Methyl 2-Acetamidopropanoate (***17k***)

Prepared according to typical procedure B to afford the product (>99% conversion, 99.1% ee with **L7**) with all characterization analysis in good accordance with the literature.

#### 3.26.1. Rhodium-Catalyzed Asymmetric Hydrogenation of α-Aryl Enamides

##### Typical Procedure C: Optimization and Substrate Scope

Optimization

Reactions were set up using a glovebox. Rh(COD)_2_OTf (0.005 mmol), ligand (0.006 mmol), *N*-(1-phenylvinyl)acetamide **18a** (0.108 g, 0.5 mmol), and the solvent (2 mL) were added to a dry 10-mL Schlenk flask containing a magnetic stir bar under an inert atmosphere. The reaction mixture was cooled under liquid nitrogen, the atmosphere was evacuated (high vacuum), and the reaction chamber was refilled with hydrogen (balloon, reactions requiring higher pressures of hydrogen were quickly transferred to an autoclave). The reaction was stirred for the designated time, filtered through a plug of Celite^®^, and washed with the solvent of choice. The solvent was removed in vacuo to yield the crude product. The ee was determined by HPLC and conversion of starting material to product by ^1^H NMR spectroscopy.

Substrate Scope

The reactions were performed with 0.5 mmol of the substrate using the procedure outlined above for the optimization process, with the following exceptions. Racemic reactions were performed with (±)-BINAP (1.2 mol %) using Rh(COD)_2_OTf (1.0 mol %), 40 bar H_2_ in THF for 18 h for substrate **18a** and Pd/C (1.0 mol %, 10 wt. % loading), 20 bar H_2_ in methanol for 0.5–2 h for **18b**–**f**. Reactions with **L1** (1.1 mol %) were performed using Rh(COD)_2_OTf (1.0 mol %), 40 bar H_2_, in CH_2_Cl_2_ for 24 h at room temperature. Reactions with **L4** (1.1 mol %) were performed using Rh(COD)_2_OTf (1.0 mol %), 20 bar H_2_, in MeOH for 1 h at room temperature. Reactions with **L7** (0.22 mol %) were performed using Rh(COD)_2_OTf (0.2 mol %), 10 bar H_2_, in THF for 1 h at room temperature except for substrate **18f**, which was subjected to Rh(COD)_2_OTf (1.0 mol %), **L7** (1.1 mol %), 60 bar H_2_, in THF for 2 h at room temperature. For the methods and chiral columns used to determine the enantiomeric excess, and chromatograms for racemic and enantioenriched products, see Appendix A and pages 15–20, respectively.

### 3.27. (S)-N-(1-Phenylethyl)acetamide (***19a***)

Prepared according to typical procedure C to afford the product (>99% conversion, 96.4% ee with **L7**) with all characterization analysis in good accordance with the literature. The absolute configuration of the product was determined by comparison of the [α]_D_^20^ value to the literature [40]. Reference value; [α]_D_^20^ = 129.5 (*c* 1.00, CHCl_3_) for the (*R*)-enantiomer (99% ee). Value obtained; [α]_D_^20^ = −52.7 (*c* 0.33, CHCl_3_).

### 3.28. (S)-N-(1-(Naphthalen-2-yl)ethyl)acetamide (***19b***)

Prepared according to typical procedure B to afford the product (>99% conversion, 97.4% ee with **L7**) with all characterization analysis in good accordance with the literature.

### 3.29. (S)-N-(1-(4-Chlorophenyl)ethyl)acetamide (***19c***)

Prepared according to typical procedure B to afford the product (>99% conversion, 97.7% ee with **L7**) with all characterization analysis in good accordance with the literature.

### 3.30. (S)-N-(1-(4-Methoxyphenyl)ethyl)acetamide (***19d***)

Prepared according to typical procedure B to afford the product (>99% conversion, 93.7% ee with **L7**) with all characterization analysis in good accordance with the literature.

### 3.31. (S)-N-(1,2,3,4-Tetrahydronaphthalen-1-yl)acetamide (***19e***)

Prepared according to typical procedure B to afford the product (23% conversion, 48.0% ee with **L7**) with all characterization analysis in good accordance with the literature.

### 3.32. (S)-Ethyl 3-Acetamido-3-Phenylpropanoate (***19f***)

Prepared according to typical procedure B to afford the product (>99% conversion, 61.3% ee with **L1**) with all characterization analysis in good accordance with the literature.

## 4. Conclusions

In summary, we have reported the design and convenient modular synthesis of a series of novel P,P-ferrocenyl pyrrolidine-containing ligands. Through-space interphosphorus coupling was observed in the ^31^P-NMR spectra for ligands L1–6, which bear (*S*)-planar chirality, indicative of a close P–P proximity in the solution phase. The potential application of the ligands was displayed in the rhodium-catalyzed asymmetric hydrogenation of dehydroamino acids and α-aryl enamides with full conversion of the starting materials and excellent ee’s observed in almost all cases using the BINOL-substituted phosphine-phosphoramidite L7. Further investigations of other catalytic asymmetric transformations are currently underway using these ligands, and progress will be reported in due course.

## Data Availability

The data presented in this study are available on request from the corresponding author.

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
