# Peer review of "Design and Synthesis of Pyrrolidinyl Ferrocene-Containing Ligands and Their Application in Highly Enantioselective Rhodium-Catalyzed Olefin Hydrogenation"

_molecules, 2022, doi:10.3390/molecules27186078_

Round 1

Reviewer 1 Report

The authors report the design and synthesis of a series of chiral pyrrolidine-substituted ferrocene-derived ligands possessing planar, central and axial chirality elements. They showed the proficiency of these ligand in the Rh-catalyzed asymmetric hydrogenation of dehydroamino acid esters and aryl enamides with full conversions and excellent levels of enantioselectivities. These ligands are interesting and holding potential for asymmetric hydrogenation for the preparation of other valuable compounds. Publication of this study in Molecules is recommended after the following concerns are well addressed.

The stability of the Pyrrolidinyl Ferrocene-containing Ligands towards air oxidation should be investigated.

Methyl-2-acetamido-3-arylacrylates 16 and N-(1-arylvinyl)acetamides 18 are the most classic and established substrates for asymmetric hydrogenation. There have been numerous ligands achieving excellent reactivities and enantioselectivities for these substrates even with remarkably low catalyst/ligand loading. The authors should at least investigate two or three other types of substrates to compare the effectiveness these newly developed ligands with those leading and privileged ones.

The absolute configuration of product 19a in table 1 is shown in the chemical structure.

The HPLC traceless given in the supporting information is not professional. The peaks should be shown in large size and high resolution, and the retention time without trace should be removed.

In both abstracts and conclusion part, is BINOL correct?

Author Response

The authors report the design and synthesis of a series of chiral pyrrolidine-substituted ferrocene-derived ligands possessing planar, central and axial chirality elements. They showed the proficiency of these ligand in the Rh-catalyzed asymmetric hydrogenation of dehydroamino acid esters and aryl enamides with full conversions and excellent levels of enantioselectivities. These ligands are interesting and holding potential for asymmetric hydrogenation for the preparation of other valuable compounds. Publication of this study in Molecules is recommended after the following concerns are well addressed.

Response: We thank this reviewer for these comments on our submission.

The stability of the Pyrrolidinyl Ferrocene-containing Ligands towards air oxidation should be investigated.

Response: We managed to obtain the X-ray structure included in this submission after prolonged (3 months) recrystallization under aerobic conditions, most likely leading to the 15% oxidation we observed. The bench stability was noted during our investigations of these ligands as being outstanding as we stored the ligands at room temperature open to the air for more than a year and they retained activity and enantioselectivity for the hydrogenations studied. This stability is similar to the BoPhoz ligands reported by Boaz.

Methyl-2-acetamido-3-arylacrylates 16 and N-(1-arylvinyl)acetamides 18 are the most classic and established substrates for asymmetric hydrogenation. There have been numerous ligands achieving excellent reactivities and enantioselectivities for these substrates even with remarkably low catalyst/ligand loading. The authors should at least investigate two or three other types of substrates to compare the effectiveness these newly developed ligands with those leading and privileged ones. 

Response: We no longer have the ligands at hand in order to perform the significant additional work suggested to extend to other substrates. However, we feel that there have been sufficient investigations performed in the present study with the Rh complexes of our novel ligands to demonstrate their efficacy and potential.

The absolute configuration of product 19a in table 2 is not shown in the chemical structure. 

Response: Yes – we have corrected this Scheme to draw the product as (S).

The HPLC traceless given in the supporting information is not professional. The peaks should be shown in large size and high resolution, and the retention time without trace should be removed. 

Response: We apologize for the quality of some of the chromatograms – these are pictures taken of printouts from the time the work was done. However, we do believe the reader can see the key points and the levels of enantioselectivities obtained.

In both abstracts and conclusion part, is BINOL correct?

Response: Yes, BINOL is most commonly used in the literature and was used consistently in the current manuscript. Binol is less frequently used.

Reviewer 2 Report

Guiry and co-workers have developed a new series of ferrocene-containing P-P ligands for asymmetric hydrogenation. The synthesis of the ligands consists of a combination of known reactions widely used for similar type of chiral ligands. However, to the best of my knowledge the present ligands have not been reported before. Then, they have also included the evaluation of the synthesized ligands in Rh-catalysed asymmetric hydrogenation of dehydroamino acid derivatives obtaining in most cases full conversion and enantioselectivities above 90%. Moreover, they have tried their ligands in asymmetric reduction of enamides with modest to excellent results.

I strongly recommend to publish the present work in molecules after addressing the following comments:

1. It is not clear what is the improvement with respect to the previously known catalysts. The authors should try some known ligands such as Josiphos or DuPhos ligands in the same reaction and compare them with their own results.

2. The crystal structure of L5 in the Fig 6 shows partial oxidation of the phosphine moiety. How can this affect the effectiveness of the ligands?  

3. Some comments about the supplementary information.

HPLC 17e. There are 3 selected peaks in the chromatogram. The retention times of the racemic and the enantioselective runs don't match.

HPLC 17f. The retention times don't match.

HPLC 17h. There are 3 selected peaks in the chromatogram. The retention times don't match. There are more than 5 minutes of difference between runs and the distance between both peaks is higher than 3 minutes between runs.

HPLC 17j. The retention times don't match. There are more than 4 minutes of difference between runs and the distance between both peaks is higher than 2 minutes between runs.

HPLC 19e. There are 3 selected peaks in the chromatogram. One of those peaks seems to be overlapping the major enantiomer.

Round 2

Reviewer 1 Report

The authors have well addressed the concerns from the reviewers. I support the publication of this paper in the present form.